# Latent Feature Mining with Large Language Models

## Abstract

Predictive modeling often faces challenges due to limited data availability and
quality, especially in domains where collected features are weakly correlated
with outcomes and where additional data collection is constrained by ethical
or practical difficulties. Traditional machine learning (ML) models struggle to
incorporate unobserved yet critical factors. We propose a framework that leverages
large language models (LLMs) to augment observed features with latent features,
enhancing the predictive power of ML models in downstream tasks. Our novel
approach transforms the latent feature mining task to a text-to-text propositional
reasoning task. We validate our framework with a case study in the criminal justice
system, a domain characterized by limited and ethically challenging data collection.
Our results show that inferred latent features align well with ground truth labels and
significantly enhance the downstream classifier. Our framework is generalizable
across various domains with minimal domain-specific customization, ensuring easy
transfer to other areas facing similar challenges in data availability.

## 1 Introduction

In numerous application domains, predicting individual outcomes and optimizing resource planning
are critical but often limited by gaps in data availability and quality. Despite the popular belief that
we operate in a "large data regime," many decisions, especially those impacting human lives, have to
be made based on small amounts of data with limited features, such as in criminal justice, healthcare,
and social services (Lu et al., 2021; Yuan et al., 2023). This poses both technical limitations and
ethical concerns. Traditional ML models, while powerful, are limited by the availability of collected
(observed) data features. This limitation is especially prominent when it comes to incorporating
unstructured data or inferring nuanced relationships between observed features and the outcomes. In
this paper, we explore how domain-informed language models can help identify latent (unobserved)
features and improve prediction accuracy for downstream tasks.

We illustrate our motivation with an example from the criminal justice setting. Predicting an
individual's in-program revocation probability (chance of committing a new crime during probation)
is critical for determining their eligibility for incarceration-diversion programs and for planning
resources like staffing ratios (Rotter and Barber-Rioja, 2015; Li et al., 2024). Typically, the data
collected includes only a limited set of features, e.g., basic demographic and criminal history
information. Crucial factors such as socio-economic status, community support availability, or
psychological profiles, which significantly influence outcomes, are often missing from these datasets.
Collecting such sensitive information can be invasive and raises ethical concerns. Additionally, the
process of gathering these data can be logistically challenging and resource-intensive. Human case
managers in these settings often have the advantage of drawing on their professional experience
and human intuition to infer these critical but unrecorded details from observed data. In contrast,
traditional ML models cannot reason beyond the explicit data provided, leading to predictions based
on incomplete information. This limitation not only undermines the accuracy of the models but also

Submitted to 38th Conference on Neural Information Processing Systems (NeurIPS 2024). Do not distribute.

poses concerns regarding the fairness of decisions derived from such data. Moreover, ML models are not designed to handle unstructured data like case notes, which may contain contextual insights to improve prediction accuracy.

Recent advancements in large language models (LLMs) offer a promising avenue to bridge these data gaps (Brown et al., 2020; Ouyang et al., 2022; Achiam et al., 2023). LLMs are capable of processing and generating information in a way that mimics human reasoning, allowing for the inference of latent features that are not directly observable but are critical for accurate predictions and decision-making. They can also analyze both structured and unstructured data to offer a holistic view of the underlying factors influencing individual outcomes.

Our proposed framework leverages LLMs to *augment observed features collected in given datasets with latent features*, enhancing the predictive power of ML models for downstream tasks such as classifications. Unlike conventional data augmentation approaches to increase the sample size, we train LLMs to infer underlying socio-economic conditions, treatment needs, and other critical but often unrecorded characteristics from collected features. This augments the feature space $X$ to improve predictions. Additionally, our framework enables generating more complete and realistic synthetic data points via learned correlations between observed and unobserved features for simulation and counterfactual policy analysis. We summarize our main contributions as follows.

1. We introduce a novel approach to formulate latent feature mining as text-to-text propositional logical reasoning. This approach effectively infers latent features from observed features, offering significantly improved accuracy and interpretability compared to alternative approaches.

2. We develop a four-step framework to implement our approach, which is generalizable with minimal domain-specific customization and has remarkably low human-annotated training data requirements. This framework expands data utility by enhancing downstream predictions without additional invasive or forbidden data collection.

3. We empirically validate our framework in the criminal justice setting to address weak observed features and unbalanced datasets. Designed as a plug-and-play solution, we demonstrate our framework's adaptability through two different prediction tasks, making it valuable for various applications with similar challenges.

## 2   Background and Related Works

**Data Augmentation and Latent Feature Extraction.**  Data augmentation is a technique commonly used in AI (Van Dyk and Meng, 2001). Generative models, such as Generative Adversarial Networks (GANs) and Variational Autoencoders (VAEs), learn data patterns and generate synthetic data to augment training sample size (Goodfellow et al., 2014; Kingma and Welling, 2013). Unlike these approaches, our framework leverages LLMs to augment the features of different individuals. Trained on crowd-sourced data rich in human behavior and societal context, LLMs have the potential to enhance feature spaces for social computing and operations improvement.

Latent features are hidden characteristics in a dataset that are not directly observed but can be inferred from available data. Incorporating meaningful latent features can enhance the performance of downstream applications (Zhai and Peng, 2016; Jiang et al., 2023). Two common approaches to infer latent features are human annotation and machine learning models. Human annotation, while reliable, is often expensive and time-consuming. It requires significant effort and resources, making it impractical for large-scale tasks. Machine learning methods like Expectation-Maximization (EM) and VAEs offer alternative techniques to infer latent features from observed data. EM algorithms estimate latent variable assignments and update model parameters to maximize data likelihood, but their results can be hard to interpret and require strong parametric assumptions. Similarly, VAEs use probabilistic approaches to describe data distribution with latent variables, but the learned mappings can also be difficult to interpret.

**Synthetic Data for Training.**  Fine-tuning is a promising approach for LLMs to reduce hallucinations and align outputs with real-world data and human preferences (Tonmoy et al., 2024; Qiao et al., 2022; Hu et al., 2021). Synthetic data has proven to be an effective, low-cost alternative to real data to improve the LLMs' reasoning performance across various domains (Liu et al., 2024). Studies by (Zelikman et al., 2022), (Wang et al., 2022) demonstrate that synthetic data improves model generalization and robustness. Our approach also uses synthetic data to augment training during fine-tuning. Unlike existing work that directly mimics observed features, we are one of the first

to formulate the generation of synthetic latent features as a reasoning task. Our approach employs few-shot prompting to create synthetic data that infers these latent features, followed by fine-tuning to enhance model accuracy and reduce hallucinations. This technique falls under the self-instruction paradigm, where models iteratively learn from augmented data.

Note that we distinguish between augmenting the feature space and augmenting training data. Our primary goal is to augment the feature space by inferring and adding latent features to the observed data to improve downstream predictions. As part of the steps in our framework to achieve this goal, we augment training data for LLM fine-tuning with synthetic samples to improve the model's reasoning capabilities.

**Incarceration-diversion Programs and Data Description.** This work conducts case studies on incarceration-diversion programs, which aim to support individuals who have committed minor offenses by providing community-based services to improve societal reintegration and reduce recidivism. Eligible individuals were diverted from traditional incarceration to such programs after risk assessment and screening. Case managers determined specific program requirements, such as substance use treatment and cognitive-behavioral therapy. There are four types of program outcomes: Completed (successfully completed the program), Revoked (committed new crimes while in the program), Not Completed (unable to finish for various reasons), and Other (unrecorded reasons).

We obtained de-identified data from our community partner for a state-wide incarceration-diversion program in Illinois. The consolidated dataset includes records of adult participants admitted to the program. The collected data features include timestamps such as the arrival and termination dates to the program, program outcomes, and individual features such as the race, gender, education, county, marriage status, housing, risk assessment scores, prior crime history, and sources of referral (e.g., from probation officer or from the court). See Appendix F for summary statistics.

# 3  The Problem Setting

In this section we formally describe our problem setting that leverages latent features to enhance downstream tasks. The downstream task we focus on is a multi-class classification problem, but the framework can easily extend to other downstream prediction tasks such as regression problems.

In a standard multi-class classification problem setting, suppose we have a dataset $D = (x_1, y_1), (x_2, y_2), \ldots, (x_n, y_n)$, where $x_i$ is a $d$-dimensional vector representing the input features $X \in \mathcal{X}$ and $y_i \in \mathcal{Y} = \{1, 2, \ldots, C\}$ denotes the corresponding class label $Y$ for individual $i = 1, \ldots, n$. The goal is to learn a classifier $f : \mathcal{X} \to \mathcal{Y}$ that accurately predicts the class labels. Consider the following scenarios in which $f$ struggles to capture the relationship between $X$ and $Y$:

1. The size of the training dataset is small relative to the complexity of the classification task or the dimensionality of the feature space;

2. When the input features $X$ are weakly correlated with class labels $Y$, the input features may not provide discriminating information to accurately predict the corresponding class labels.

To address these challenges, we could use additional informative features to enhance the classifier's ability to capture the relationship between $X$ and $Y$. Latent features can serve such a purpose.

**Definition of Latent Features.**
Latent features, denoted as $Z$, represent underlying attributes that are not directly observed within the dataset but are correlated with both the observed features $X$ and the class labels $Y$. We use a function $g$ with $Z = g(X)$ to denote the correlations between the latent features and the observed features $X$. As shown in figure 3, latent features $Z$ are correlated with $X$ and $Y$. One can learn the latent features from the original features $X$ and augment the features $f(\mathbf{X}, \mathbf{Z})$ to learn the classifier $Y$.

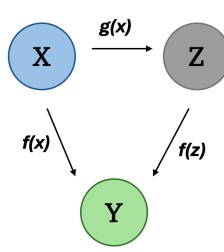

In typical ML settings, latent features primarily reduce the dimensionality of the feature space. Beyond this, latent features can capture discriminative information not explicitly present in the original features. Our approach focuses on this latter benefit, extracting informative latent representations to help classifiers better differentiate between classes. Essentially, $Z$ acts as ensemble features derived from the original features $X$, capturing complex patterns that individual features might miss, especially when $X$ is weakly correlated with the outcome $Y$.

While this approach seems beneficial intuitively, it is important to note that adding more features is not always helpful if the extracted features are not meaningful and introduce noise. In the following lemma, we show in a simple logistic regression setting that while adding features can reduce in-sample loss, it does not always reduce out-of-sample loss if the added features are not informative. We use the log-loss (the cross-entropy loss) of the logistics regression for binary outcome $Y \in \{0, 1\}$. We denote the optimal coefficients that minimize the in-sample log-loss function as $\beta^*$ for the original features and $\tilde{\beta}^*$ for the augmented features.

**Lemma 1.** *The in-sample log-loss always follows* $\mathcal{L}^{in}(\tilde{D}, \tilde{\beta}^*) \leq \mathcal{L}^{in}(D, \beta^*)$. *When the added features are non-informative, there exist instances such that the out-of-sample log-loss* $\mathcal{L}^{out}(\tilde{D}, \tilde{\beta}^*) > \mathcal{L}^{out}(D, \beta^*)$.

The results in the lemma can be generalized to multi-class labels. Since augmenting the feature space is not necessarily beneficial unless the added features are meaningful, a major part of our case study is to empirically test whether the extracted features from our framework indeed improve downstream prediction. If the added features significantly enhance downstream prediction accuracy, this provides strong evidence that the inferred latent features are meaningful.

# 4 Latent Feature Mining with LLMs

To overcome the limitations of existing approaches, we propose a new approach to efficiently and accurately extract latent features and augment observed features to enhance the prediction accuracy. At a high level, our approach transform the latent feature mining as a text-to-text propositional reasoning task, i.e., infer the relationship $Z = g(X)$ through logical reasoning with natural language.

Following the framework established in previous work (Zhang et al., 2022), we denote the predicates related to the observed features as $P_1, P_2, \ldots, P_m$. Consider a propositional theory $S$ that contains rules that connect $P$'s to the latent feature $Z$. We say $Z$ can be deduced from $S$ if the logic implication $(P_1 \wedge P_2 \wedge \ldots \wedge P_m) \rightarrow Z$ is covered in $S$. For potentially complicated logical connections between $P$'s and $Z$, we also introduce intermediate predicates $O$'s and formulate a logical chain (a sequence of logical implications) that connects $X$ to the latent features $Z$ as follows:

$$X \rightarrow (P_1 \wedge P_2 \wedge \ldots \wedge P_m) \rightarrow (O_1 \wedge O_2 \wedge \ldots \wedge O_\ell) \rightarrow Z. \tag{1}$$

Our approach formulates this logical chain as a multi-stage Chain of Thoughts (CoT) prompt template, and then guide LLMs to infer $Z$ from $X$ using the prompt template. Specifically, we first extract predicates $P$'s from $X$. Then we infer intermediate predicates with a rule $(P_1 \wedge P_2 \wedge \ldots \wedge P_m) \rightarrow O_l$ for $l = 1, \ldots, \ell - 1$, and forward the intermediate predicates into the next stage to infer $O_{l+1}$. Finally, we infer latent features with $(O_1 \wedge O_2 \wedge \ldots \wedge O_\ell) \rightarrow Z$. With the formulated multi-stage CoT prompt template, we generate synthetic data to fine-tune LLMs to enhance the logical reasoning ability of LLMs in self-instruct fashion (Wang et al., 2022), and ensure that the generate text is aligned with each step of our desired "chain of reasoning" format.

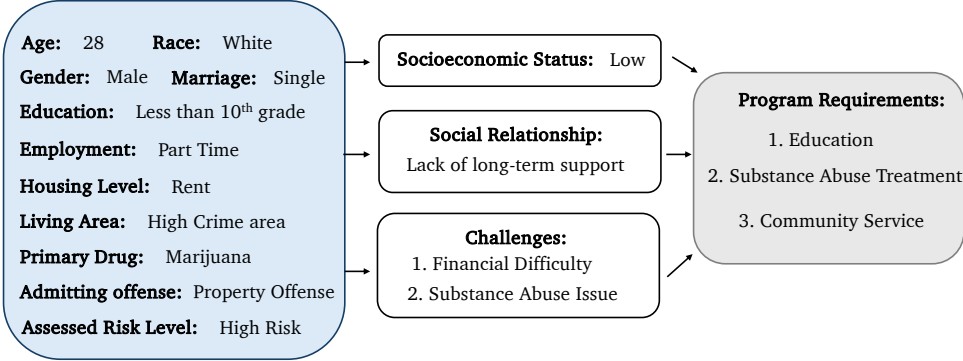

Figure 1: Example of latent feature mining through chain of reasoning

We use a hypothetical example from our case study setting to illustrate the formulation of the logic chain. The blue (leftmost) box in Figure 1 shows the observed feature $X$ for one individual. Examples for the predicates $P$'s formulated from $X$ could be:

$P_1$ :*"the client has part-time job"*, $P_2$ : *" the client hasn't complete high school"*,
$P_3$ :*"the client is single"*, $P_4$ : *"the client has drug issue"*, $P_5$ :*" the client lives in high crime area"*, $P_6$ : *" the client is assessed with high risk"* ...

To infer the latent feature $Z$ – in this example, the required programs to attend during probation – we go through a multi-stage reasoning to infer the intermediate predicates $O$'s; see the white (middle) boxes in Figure 1. One example logic that connects $P$'s to $O$'s could be:

$P_1$ = *"The client has unstable employment"*
$P_2$ = *"The highest education level of client is less than 10th grade"*
$O_1$ = *"The client has low socioeconomic status"*
If $(P_1 \wedge P_2 \rightarrow O_1) \in S$, then $O_1$ is True.

Finally, with $P$'s and $O$'s, we can connect $X$ with $Z$ though the logic chains. One example of the logical chain is as follows:

> *"The client is grappling with unstable employment and a relatively low educational level, factors that likely contribute to a low socioeconomic status. Additionally, being single, struggling with drug issues, and residing in a high-crime area further exacerbate the lack of positive social support. Given these circumstances, education could serve as a valuable intervention. Community service can be particularly beneficial for someone who is single and may lack a broad support network. Substance abuse treatment is crucial for individuals from lower socioeconomic backgrounds to aid in recovery from substance abuse. Hence we can choose education, substance abuse treatment, community service for this client."*

Here, *"unstable employment and a relatively low educational level"* and *"being single, struggling with drug issues, and residing in a high-crime area"* are $P$'s extracted from the features $X$, while *"a low socioeconomic status"* and *"lack of positive social support"* are $O$'s. Finally, the rationales *"education could serve as a valuable intervention . . . recovery from substance abuse. Hence we can choose education, substance abuse treatment, community service for this client* connect the intermediate predicates to the latent variables $Z$ (program requirements) we want to infer, i.e., $Z_1$=‘education’, $Z_2$=‘substance abuse treatment’, $Z_3$=‘community service’.

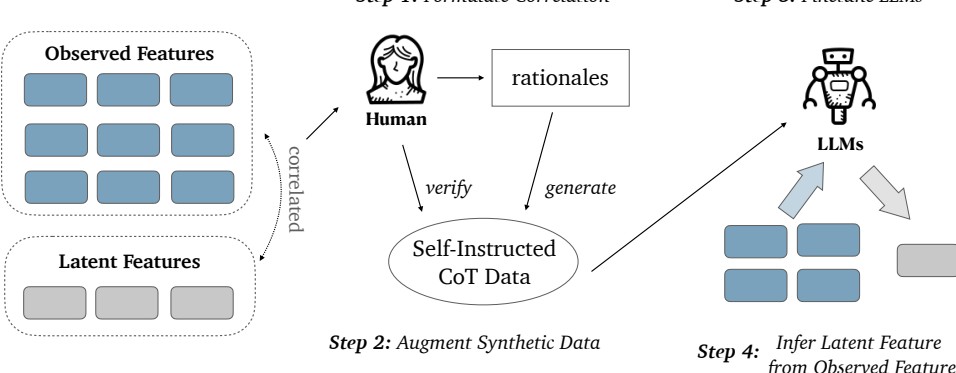

Figure 2: Overview of latent feature inference framework.

Figure 2 illustrates the full process of of our proposed framework with four steps.

**(1) Formulate baseline rationales:** The first step is to formulate baseline rationales, whic serve as guidelines for LLMs to infer latent features from observed ones. This involves two sub-steps:

– The first sub-step is to develop some baseline rationales, i.e., identify observed features potentially correlated with latent features and formulate their relationships – the logic chain that connects $X$ to $Z$. Sources to help formulate these baseline rationales include established correlations (e.g., risk score formulas), human input, and external information like socio-economic status in the neighborhood.

– In the second sub-step, we craft prompts with interactive alignment. This is a critical component to establish correct reasoning steps for prompts used in Step 2 to generate synthetic rationales. We

involve human who are experienced in the domain to provide a prompt template for LLMs to generate rationales aligned with the baseline rationales, then test the prompt template on a few examples using zero-shot. If the LLM fails to certain example, we provide the ground truth back to the LLM, allowing it to revise the prompt template (Miao et al., 2023). This process iteratively refines the template until LLMs consistently generate the desired output for all selected examples.

**(2) Enlarge data with synthetic rationales for fine-tuning:** We generate synthetic training data in self-instruct fashion (Wang et al., 2022). With a handful of examples of the baseline rationales as a reference, we then guide the LLMs via in-context learning to generate similar rationales to enlarge the training data samples. To ensure the quality and diversity of the generated dataset, we introduce human-in-the-loop interventions to filter out low-quality or invalid data based on heuristics. We also leverage automatic evaluation metrics for quality control, e.g., removing data that lack essential keywords.

**(3) Fine-tuning LLMs:** To enhance the reasoning capabilities of the LLMs and better align their outputs in specific domains, we employ a fine-tuning process which utilizes the processed dataset from the previous step (Qiao et al., 2022). Fine-tuning not only boosts the accuracy and reliability of the LLMs, but also significantly improves their ability to reason with complex inputs, and reducing hallucination (Tonmoy et al., 2024).

**(4) Latent feature inference:** The fine-tuned model is able to mirror the nuanced decision-making process of human experts. We use the fine-tuned model to identify latent features and feed them into downstream prediction tasks.

Regarding the generalizability of our framework, Steps 2-4 rely primarily on the mechanics of LLMs, which naturally have a high degree of adaptability across different domains. Step 1, which involves the identification and formulation of baseline domain-specific rationales, requires more expert knowledge. To assist with Step 1, our interactive-alignment strategy can help craft effective prompts by allowing iterative refinement based on feedback, reducing the burden on domain experts.

## 5 Experiments Setup

In this section, we demonstrate the efficacy of our proposed framework on a unique dataset from a state-wide incarceration diversion program as described in Section 2. We design two sets of experiments to empirically investigate: (1) Can our approach accurately imitate the human thinking process to infer latent features? (2) Is our approach more effective than alternative techniques to infer latent features? (3) Does our approach enhance the performance of downstream prediction tasks?

In the first experiment, we treat the risk level of individuals as a latent feature, despite it being collected in the dataset. This experiment examines whether the latent features $\hat{Z}$ inferred by LLMs match well with the actual features $Z$. In the second experiment, we assume that the program requirements are latent features, which lack ground truth labels for most individuals (only a few dozen individuals have the program requirements recorded in the data). We first have LLMs deduce these requirements, then add them to the downstream prediction task of program outcomes $Y \sim f(X, \hat{Z})$ and evaluate whether the prediction accuracy is improved, i.e., the inferred features are indeed beneficial and not detrimental (recall the results in Lemma 1).

### 5.1 Risk Level Prediction

**Task Description.** In this task we treat an observed feature—Risk Level—as the latent feature to infer. The task is a multi-classification problem to learn $Z \sim g(X)$ among four labels for the latent variable $Z \in \{moderate, high, very\_high\}$ based on each client's profile $X$.

**Implementation Details.** We implement our proposed framework as follows. All prompt templates are attached to Appendix section C.

- Step 0. Profile writing: In this pre-processing step, we translate structured profile data $X$ into text that can be better handled by LLMs, i.e., formulating predicates $P$'s from the features $X$. To enrich the profile with important in formations that could potential benefits the following steps, we formulate the intermediate predicates $O$'s, where we prompt LLMs to extract and summarize underlying information such as background, socio-economic status, and challenges in two or three sentences. We then merge these sentences into the client's profile. We use zero-shot prompting with GPT-4 for this step.

- Step 1. Formulating rationales: Using human input, established risk score calculations (Corrections), and the code book with risk calculation details provided by our community partner, we summarize a general rule for inferring risk levels from the predicates, i.e., establishing the logic chains from $P$'s and $O$'s to $Z$. We then sample 40 client features from the dataset and manually formulate 40 baseline rationales that logically connect features to corresponding risk levels and that are aligned with the high-level general rule. To avoid the primacy effect of LLMs, we rate risk scores from 0 to 10 to add variability in the labels, categorized as follows: 0-4 (moderate risk), 4-7.5 (high risk), and 7.5-10 (very high risk).

- Step 2. Enlarge fine-tuning data: With the 40 baseline rationales, we generate additional synthetic rationales. We sample client features and corresponding ground truth risk scores from the dataset, using one of the 40 rationales as an example, to prompt LLMs to produce similar narratives with CoT prompts. In total we got 3000 rationales for the training data.

- Step 3. Fine-tune LLMs: Our framework is designed to be plug-and-play, allowing the synthetic data generated in the previous step to be used across different language models. We fine-tune two pre-trained language models for cross-validation purposes: GPT-3.5 and Llama2-13b(OpenAI, 2021). We use OpenAI API to fine-tune GPT-3.5-turbo-0125 (Touvron et al., 2023; OpenAI). We fine-tune Llama2-13b-chat using LoRA (Hu et al., 2021).

- Step 4. Inference with LLMs: We prompt fine-tuned LLMs to infer risk level $\hat{Z}_i$ from features $X_i$ for each client $i$ in the test data and evaluate the out-of-sample accuracy by comparing the inferred latent variable (risk level) $\hat{Z}_i$ with the ground truth label $Z_i$.

**Evaluation.** We choose ML classifiers (e.g., Neural Networks or Gradient Boosting Trees) as the baseline to infer $\hat{Z}_i$ from features $X_i$. We compare the prediction performance of $\hat{Z}_i$ inferred from our approach with that from ML models using out-of-sample accuracy and F1 score. Additionally, we evaluate the quality of generated text with an automatic evaluation metric. In the pre-processing step, we assess the keyword coverage rate in the generated profile assuming each feature value is a keyword. For synthetic rationales, we use YAKE, a pretrained keyword extractor (Campos et al., 2020), to identify keywords. We then evaluate the keyword coverage rate with a rule-based detector to determine how many logical information points are covered.

### 5.2 Outcome Prediction

**Task Description.** In this task, we treat the program requirements (e.g., substance treatment, counseling) for each client as the latent features $Z$ and use them to augment the original feature $X$ for outcome prediction, which is a multi-classification problem to learn $Y \sim f(X, Z)$ among four labels for the outcome $Y \in \{Completed, Revoked, NotCompleted, Other\}$. The raw dataset does not record the program requirements except for a very few clients; thus, the latent feature $Z$ in this task is truly unobservable (in contrast to the one used in the first task). Available program requirement options for this task are attached to the appendix section D.

**Implementation Details.** Steps 0 and 2-4 remain almost the same as in the risk-level prediction task. Step 1 requires a slight adjustment (as discussed in Section 4, this step is the main part in our framework that requires customization). Here, we formulate 40 baseline rationales in step 1 to deduce clients' program requirements from their features. We leverage multi-stage prompting strategy (Qiao et al., 2022) to break down the task into three sub-tasks: (1) identify the main challenges from the client's profile, (2) rank these challenges by priority, (3) match the challenges with suitable requirements. Particularly, the third task is our main goal, with the first two serving as steps to streamline the process and simplified the task.

**Evaluation.** We train an ML classifier to predict outcomes with and without the inferred latent features, i.e., $\hat{Y}_i \sim f(X_i, \hat{Z}_i)$ versus $\hat{Y}_i \sim f(X_i)$. We evaluate the out-of-sample accuracy by comparing the predicted outcome $\hat{Y}_i$ with the true label $Y_i$ in the test data. This comparison allows us to assess whether incorporating the latent features enhances the classifier's performance.

## 6 Results

In this section, we demonstrate experiments results for two case studies we designed and additional results for sensitivity analyses.

## 6.1 Risk Level Prediction Results

As mentioned in Section 5.1, we infer risk level on the client's profile. We compare our approach's performance to baseline ML model's performance using the accuracy score and F1 score. Before showing this performance comparison, we first show results on the generated text quality.

**Generated Text Quality.** For profile writing in Step 0, we treat each individual feature in $X_i$ as a keyword to cover, and measure the keyword coverage rate. The generated profiles demonstrated an average keyword coverage rate of 98%, indicating that they effectively capture the most important information from the original data. For the generated synthetic rationales in Step 2, we treat terms such as age, gender, employment, and education as critical keywords and assess their coverage rate. The fine-tuned GPT-3.5 and Llama2-13b-chat both achieved a keyword coverage rate of 100%. This indicates that the generated content adheres strictly to the guidelines established in the training data, ensuring that all necessary information is accurately represented.

**Latent Variable Inference Performance.** As shown in Figure 3(a), our approach achieves the highest overall accuracy. In particular, the fine-tuned GPT-3.5 achieves an accuracy that is 20% higher than other baseline ML approaches. The reason that ML models struggle to predict well is due to the fact that there is no strong correlation between the observed features and the targets (risk level); see the correlation plot in Appendix F. In contrast, our approach demonstrates superior performance, since it more effectively handles datasets with subtle or non-obvious relationships between the observed and target variables. This result shows that **our approach is able to make accurate inference of latent features and outperforms traditional ML approaches**.

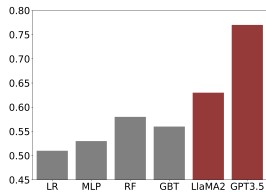

(a) Model accuracy

| Category | LR | MLP | RF | GBT | LLaMA2 | GPT3.5 |
|----------|------|------|------|------|--------|--------|
| Moderate | 0.51 | 0.54 | 0.44 | 0.46 | 0.57 | **0.69** |
| High | 0.65 | 0.55 | 0.69 | 0.66 | 0.70 | **0.81** |
| Very High | 0.20 | 0.11 | 0.18 | 0.18 | 0.38 | **0.81** |

(b) F1 scores

Figure 3: Risk level prediction results: (a) Model accuracy; (b) F1 scores per-category. LR - logistic regression; MLP - Neural Networks; RF- random forest; GBT - Gradient Boosting Trees.

Table 3(b) details the prediction performance by class, showing F1 scores for each class using ML models and our approach. Notably, all ML models struggle with the 'Very High Risk' category – this category is often misclassified as 'High Risk' due to similar feature distributions of these two categories and unbalanced data (only 371 training points for 'Very High Risk'). In contrast, our approach significantly improves the prediction performance for this category, highlighting its **effectiveness for unbalanced datasets.** This improvement is likely because our LLM-based approach has intermediate steps (profile writing to obtain the socio-economic status and other contextual factors in step 0 and connecting these factors with the latent variables in step 1), which help capturing the subtle distinctions between 'High Risk' and 'Very High Risk' that are not explicitly recorded.

## 6.2 Outcome Prediction Results

As mentioned in Section 5.2, we infer program requirements as additional latent features and use them for the downstream outcome prediction task. We compare the performance of the downstream classifiers that trained with and without the latent features. Note that in the first task (risk-level inference), GPT3.5 demonstrated better performance than llama2-13b. Thus, we focused on fine-tuning GPT-3.5 when using our approach for this task.

As illustrated in Table 4(a), incorporating latent features significantly improves the performance of the downstream classifiers. Specifically, the addition of latent features increases the ROC AUC score of Logistic Regression from 0.70 to 0.89 and from 0.84 to 0.92 for the Gradient Boosting Tree. Furthermore, the feature importance in Figure 4(b) shows that the inferred features – 'requirement_1', 'requirement_2', and 'requirement_3' – are among the top-ranked features. This implies the significant relevance of these features on the downstream classification task. Hence, we can conclude that **our approach has the capability of enhancing the downstream classifier's accuracy with inferred latent features**.

| without latent feature | LR | MLP | GBT |
|:---:|:---:|:---:|:---:|
| ROC AUC Score | 0.70 | 0.81 | 0.84 |
| F1 Score | 0.69 | 0.70 | 0.71 |

| with latent feature | LR | MLP | GBT |
|:---:|:---:|:---:|:---:|
| ROC AUC Score | **0.89** | **0.88** | **0.92** |
| F1 Score | 0.75 | 0.73 | 0.77 |

(a) Model Performance

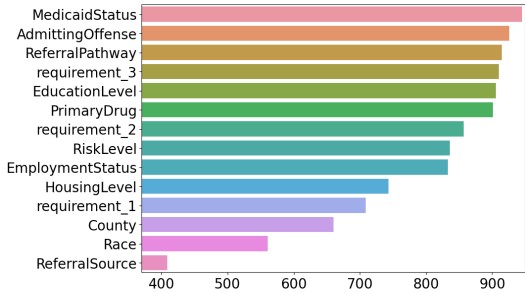

(b) Feature Importance Plot

Figure 4: Outcome prediction results: (a) Model performance with/without the inferred latent features (program requirements); (b) feature importance plot. LR - logistic regression; MLP - Neural Networks; GBT - Gradient Boosting Trees.

## 6.3 Sensitivity Analysis

In our sensitivity analysis, we further investigate the following three questions: (1) How sensitive is our approach to the quality of human guidelines? (2) How important is fine-tuning in our framework?

For the first question, perhaps not surprisingly, our approach is sensitive to human guidelines, specifically the baseline rationales and prompt templates formulated in Step 1. We have conducted an ablation study to determine the optimal level of details required in the prompts. As shown in Figure 9 in Appendix D, the best performance was achieved with the most reasoning steps and a sentence length of two per step. In other words, increasing the number of reasoning steps allows us to decompose the task into simpler components and enhances the performance of LLMs. More importantly, while human guidelines are important, **the interactive self-revise alignment strategy can significantly help** during the sub-step of Step 1 (prompt crafting). By providing ground truth and encouraging self-reflection, GPT-4 can revise the prompt template to include crucial details, ensuring a more accurate evaluation.

The answer to the second question is that **fine-tuning is necessary**. We have conducted another ablation study, where we repeated the risk-level prediction task with zero-shot, one-shot, and three-shot prompting to compare with our fine-tuned model. In zero-shot, we provided only the task description. In one-shot and three-shot, we included randomly selected human-verified examples. Accuracy rankings from lowest to highest were: three-shot (40%), zero-shot (55%), one-shot (60%), and the fine-tuned model (75%); see Table 9 in Appendix D. The three-shot's poor performance may be due to information loss from long inputs. Zero-shot responses are highly variable and not well-suited for downstream tasks. Although one-shot showed improvement, the fine-tuned model significantly outperformed all others.

## 7 Discussion

This study presents a framework that leverages the capabilities of LLMs to enhance the prediction accuracy in downstream tasks without necessitating invasive data collection methods. Our approach reduces the need for collecting extensive personal data, thus mitigating privacy concerns. This aligns with ethical data usage standards, especially in sensitive domains. Note that we do not explicitly address bias in the data or LLM reasoning processes in this paper. We excluded the 'race' feature in our case study and found alignment in risk level distributions across genders, implying no additional bias introduced by our approach. However, existing biases in LLMs could be perpetuated if not monitored and adjusted. Addressing these biases is beyond this paper's scope and is left for future research as a critical area.

This framework has vast potential applications, particularly in areas with limited data and ethical constraints. For example, in healthcare, our framework can help predict readmission or post-discharge mortality by inferring unrecorded social determinants of health. For low-volume niche product recommendations, our framework can synthesize customer preference data to enhance recommendation systems without extensive user tracking.

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

# Appendix

## A  Proof of Lemma 1

468 We use the log-loss, defined as

$$\mathcal{L}(D, \beta) = -\frac{1}{n} \sum_{i=1}^{n} [y_i \log(p_i) + (1 - y_i) \log(1 - p_i)] \tag{2}$$

469 for given data $D = \{(x_i, y_i)\}_{i=1}^{n}$ and $p_i = 1/(1 + e^{-(\beta_0 + \beta_1 x_i)})$. When using the augmented feature
470 $\tilde{x}_i = (x_i, z_i)$, we denote the data as $\tilde{D} = \{((x_i, z_i), y_i)\}_{i=1}^{n}$.

471 For the first part of the lemma, we note that the in-sample log-loss for the original features follows

$$\mathcal{L}^{\text{in}}(D, \beta) = -\frac{1}{n} \sum_{i=1}^{n} [y_i \log(p_i) + (1 - y_i) \log(1 - p_i)], \tag{3}$$

472 and the in-sample log-loss for the augmented features follows

$$\mathcal{L}^{\text{in}}(\tilde{D}, \beta) = -\frac{1}{n} \sum_{i=1}^{n} [y_i \log(\tilde{p}_i) + (1 - y_i) \log(1 - \tilde{p}_i)], \tag{4}$$

473 where $p_i = 1/(1 + e^{-(\beta_0 + \beta_1 x_i)})$ and $\tilde{p}_i = 1/(1 + e^{-(\beta_0 + \beta_1 x_i + \beta_2 z_i)})$.

474 We denote the optimal coefficients that minimize the log-loss in (3) as $\beta^* = (\beta_0^*, \beta_1^*)$, and the
475 coefficients that minimize the log-loss in (4) as $\tilde{\beta}^* = (\tilde{\beta}_0^*, \tilde{\beta}_1^*, \tilde{\beta}_2^*)$. Note that $\check{\beta} = (\beta_0^*, \beta_1^*, 0)$ is a
476 feasible solution for the log-loss in (4). Therefore, using the optimization property, we have

$$\mathcal{L}^{\text{in}}(\tilde{D}, \tilde{\beta}^*) \leq \mathcal{L}^{\text{in}}(\tilde{D}, \check{\beta}) = \mathcal{L}^{\text{in}}(D, \beta^*),$$

477 which completes the first part of the lemma.

478 For the second part of the lemma, we first assume that for the given data $D$, $\mathcal{L}^{\text{in}}(\tilde{D}, \tilde{\beta}^*) =$
479 $\mathcal{L}^{\text{in}}(D, \beta^*) - \epsilon/n$ where $\epsilon \geq 0$ from the first part of the lemma. We now construct an instance
480 with an out-of-sample dataset $D'$ that contains $n + 1$ samples, where $D'$ consists of (i) the $n$ data
481 points that exactly match with $D$ (or $\tilde{D}$) for the first $n$ samples, and (ii) one additional sample
482 $(x_{i+1}, y_{i+1})$ (or $((x_{i+1}, z_{i+1}), y_{i+1})$ when using the augmented features). Without loss of generality,
483 assume that $y_{i+1} = 1$. Then we have

$$\mathcal{L}^{\text{out}}(D', \beta^*) = \frac{1}{n+1} \left( n\mathcal{L}^{\text{in}}(D, \beta^*) - \log(p_{i+1}) \right)$$

484 and

$$\mathcal{L}^{\text{out}}(\tilde{D}', \tilde{\beta}^*) = \frac{1}{n+1} \left( n\mathcal{L}^{\text{in}}(\tilde{D}, \tilde{\beta}^*) - \log(\tilde{p}_{i+1}) \right).$$

485 When the added features $Z$'s are non-informative, we consider the scenarios that they are noise and
486 the additional term $\tilde{\beta}_2^* Z$ also contributes noise to the predictions. In other words, the coefficients $\tilde{\beta}^*$
487 do not generalize well to the test data. Therefore, there exists an instance where the realization of $Z$,
488 $z_{i+1}$ deviates from the predicted probability significantly, such that

$$\tilde{p}_{i+1} < p_{i+1}/\exp(\epsilon) \leq p_{i+1}.$$

489 Note that this instance exists since the noise terms do not correspond to any actual pattern in the test
490 data, causing incorrect predictions, and in our construction, a smaller predicted probability would be
491 less accurate as the label $y_{i+1} = 1$. Therefore,

$$-\log(\tilde{p}_{i+1}) > -\log(p_{i+1}) + \epsilon,$$

492 and

$$
\begin{aligned}
\mathcal{L}^{\text{out}}(\tilde{D}', \tilde{\beta}^*) &= \frac{1}{n+1} \left( n\mathcal{L}^{\text{in}}(D, \beta^*) - \epsilon - \log(\tilde{p}_{i+1}) \right) \\
&> \frac{1}{n+1} \left( n\mathcal{L}^{\text{in}}(D, \beta^*) - \log(p_{i+1}) \right) = \mathcal{L}^{\text{out}}(D', \beta^*).
\end{aligned}
$$

## B  Compute Resources

494 For all experiments, we split data into training and testing dataset with ratio of 8:2.

495 For experiment 1 (risk level prediction), we finetune LLaMA2-13b-chat on 2 X NVIDIA RTX A6000
496 for 4 hours with LoRA. And we finetuned three times for different subtasks. We use OpenAI offical
497 API to finetune GPT3.5 model, which requires no GPUs. Each finetune job takes about 2 hours. We
498 repeat 3 times for different sub tasks. Additionally, we also run Machine Learning baseline model on
499 CPU (Intel i7). We run grid search for each classifier.

500 For experiment 2 (outcome prediction), we use OpenAI offical API to finetune GPT3.5 model, which
501 requires no GPUs. Each finetune job takes about 2 hours. We repeat 6 times for different sub
502 tasks.Additionally, we also run Machine Learning baseline model on CPU (Intel i7). We run grid
503 search for each classifier.

504 All other experiments (e.g. sensitive experiment) are conducted on ChatGPT, which requires no GPU.

## C  Prompt template

```
Task: Write a paragraph to profile the client, please include following:

1. Write sentences to cover all basic information provided.
2. Provide information about the area of this client live in, as much more details as you can.
3. Infer social economic status of this client
4. Infer the challenges that this client might facing.

Here are the basic information of the client: <features>.

Here is the reference of living area context: <additional info>
```

Figure 5: Profile writing prompt

```
Here is the profile of a client: <profile>
Given the client's information, please infer a risk score out of 10.

Given client's information to infer risk score out of 10, we know that:
1. Employment (If client has unstable employment status, increase the score by 1.
Adjust score if needed):  ___
2. Financial Status (If client has financial difficulty, increase the risk score by 1.
If client relies on social economic assistance, further increase the risk score by 1.
Adjust score if needed.): ___
3. Education (Increase the risk score by 1 if the highest grade of school completed is
less than grade 12. Further increase the risk score by 1 if the highest grade completed
is less than grade 10): ___
4. Family and Marital (Increase score if client is dissatisfied with his/her current
marital relationships situation. Increase risk score if the client is a social isolate.
Adjust score if needed.):  ___
5. Drug (Increase risk score by 1 if the client has ever had a drug problem. If the
drug problem is related with Heroin, further increase the risk score by 1. Adjust score
if needed.):  ___
6. Living Area (Increase risk score by 1 if the client lives in a high crime
neighborhood): ___
7. Age (Increase risk score by 0.3 if the client is under the age of sixteen):
8. Gender (Increase risk score by 0.3 if the client is male):
Conclusion: ___
```

Figure 6: Risk Level Prediction: Prompt template and response CoT template

```
Here is the profile of a client: <profile>
Analyze the provided profile of the client to infer the main challenges he faces.
```

```
Given the identified challenges for the client, infer the priority of each
challenge in terms of immediate action and long-term impact on his reintegration
into society. Please response in the ranking order. Here are the challenges: Here
are the challenges <challenges>:
```

```
Here is the available list of programs <program list>:
Given the profile and challenges of the client, select the top 3 program
requirements that would be most beneficial for the client.
Here is the profile of client: <profile + top 3 ordered challenges>
```

Figure 7: Requirement selection: Multi-stage Prompt template

```
To select the top 3 programs that would be most beneficial for the client, let's analyze each
available options:
1. Thinking for a Change (It aims to transform criminogenic thinking patterns with designed
cognitive-behavioral curriculum. Recommend for clients assessed at relatively high risk
level): __
2. Employment (It aims to help client develop employability. Recommend this for clients with
unstable employment status): __
3. Education (It aims to engage clients in educational programs. Recommend clients without a
high school diploma or GED):__
4. Positive Peer Mentoring (It offers positive role models and fosters a supportive network,
which can deter criminal associations. Recommend this for clients residing in high-crime
areas):__
5. Community Service (It aids in building a sense of responsibility and community connection.
Recommend for clients with property offense or drug-related offenses):__
6. Mental Health Treatment (It addresses underlying mental health issues that may contribute
to criminal behavior. Recommend for clients with a history of substance abuse or unstable
living conditions):__
7. Anger Management (It focuses on teaching effective emotion and reaction management
techniques. Recommend for clients who exhibit aggressive behaviors or have property-related
offenses):__
8. Substance Abuse Treatment (It aims to help clients overcome substance dependencies.
Recommend for clients with histories of drug-related offenses or primary drug use):__
9. Domestic Violence Counseling (It aims to address and modify violent behavior patterns.
Recommend for clients involved in violent incidents):__
10. Sex Offender Counseling (It focuses on behavior modification and preventing recidivism.
Recommend for clients with sex-related offenses):__
Conclusion: ___
```

Figure 8: Requirement selection: Response CoT template

# D    Ablation Study Results

| Setting | Accuracy |
|---------|----------|
| Zero-shot | 55% |
| One-shot | 60% |
| Three-shot | 40% |
| Fine-tune | 75% |

(a) Risk level prediction results across different setting

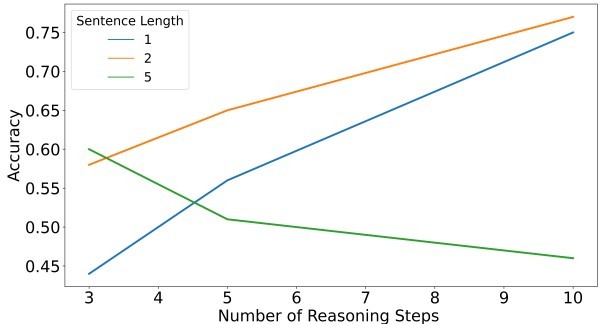

(b) Risk level prediction results across different strategy

Figure 9: Ablation study results: (a) Experiments on risk level prediction task using GPT4 with different prompting setting. (b) Experiments using GPT4 with different prompting setting different prompting strategies.

# E    Program Requirements

| Requirement Name | Description |
|------------------|-------------|
| Thinking for a Change | Aimed at transforming criminogenic thinking patterns using a cognitive-behavioral curriculum, recommended for clients at a high risk level. |
| Employment | Helps develop employability, recommended for clients with unstable employment status. |
| Education | Engages clients in educational programs, recommended for those without a high school diploma or GED. |
| Positive Peer Mentoring | Provides positive role models and a supportive network, recommended for clients in high-crime areas. |
| Community Service | Builds a sense of responsibility and community connection, recommended for clients with property or drug-related offenses. |
| Mental Health Treatment | Addresses underlying mental health issues, recommended for clients with a history of substance abuse or unstable living conditions. |
| Anger Management | Teaches emotion and reaction management techniques, recommended for clients who exhibit aggressive behaviors or have property-related offenses. |
| Substance Abuse Treatment | Helps overcome substance dependencies, recommended for clients with drug-related offenses or primary drug use. |
| Domestic Violence Counseling | Addresses and modifies violent behavior patterns, recommended for clients involved in violent incidents. |
| Sex Offender Counseling | Focuses on behavior modification and preventing recidivism, recommended for clients with sex-related offenses. |

Table 1: Available Programs

 # F  Data Description

Table 2: Categorical Covariates Summary Statistics (N/A or Other Categories are Omitted).

| Variable | Categories | County | | | |
| --- | --- | --- | --- | --- | --- |
| | | DuPage | Cook | Will | Peoria |
| Risk | Highest | 24.3 | 32.0 | 2.3 | 1.0 |
| | High | 60.7 | 26.2 | 35.1 | 24.7 |
| | Medium | 11.0 | 15.6 | 42.1 | 47.0 |
| AdOffense | Drugs | 43.0 | 67.8 | 31.7 | 37.0 |
| | Property | 31.1 | 17.6 | 52.5 | 46.3 |
| | DUI | 11.1 | 2.3 | 3.8 | 1.0 |
| OffenseClass | Class 4 | 42.5 | – | 11.5 | 20.6 |
| | Class 3 | 13.5 | – | 5.7 | 5.7 |
| | Class 2 | 16.0 | – | 5.7 | 5.1 |
| Pdrug | Heroin | 27.0 | 43.6 | 32.3 | 9.5 |
| | THC | 18.6 | 18.5 | 17.5 | 21.6 |
| | Coc.Crack | 7.8 | 10.9 | 21.0 | 11.6 |
| ReferralReason | Tech Violation | 31.2 | 0.0 | 12.8 | 0.0 |
| | 3/4 Felon | 20.5 | 70.5 | 59.2 | 80.0 |
| | 1/2 Felon | 9.8 | 16.5 | 23.7 | 14.7 |
| WhoReferred | Prob Officer | 64.7 | 97.3 | 1.8 | 0.0 |
| | Judge | 32.0 | 1.3 | 0.7 | 91.3 |
| | Pub. Defender | 0.6 | 0.0 | 75.3 | 2.8 |
| Gender | Female | 25.2 | 21.3 | 21.7 | 19.8 |
| | Male | 74.8 | 77.5 | 78.2 | 80.0 |
| EmplymntS | Full Time | 49.7 | 85.7 | 38.2 | 6.7 |
| | None | 32.3 | 4.8 | 59.2 | 92.0 |
| | Part Time | 18.0 | 9.4 | 2.7 | 1.3 |
| MaritalS | Single | 86.4 | 85.6 | 15.0 | 22.9 |
| | Married | 5.9 | 7.1 | 1.8 | 5.7 |
| | Divorced | 4.7 | 2.3 | 0.2 | 1.8 |
| EducationS | HighSchool | 40.3 | 37.2 | 34.3 | 13.6 |
| | No HighSchool | 32.6 | 52.4 | 10.8 | 12.3 |
| | Some College or Graduated | 19.4 | 3.5 | 11.8 | 4.4 |
| HousingS | Friend or Family | 62.3 | 27.9 | 6.2 | 17.7 |
| | Own/Rent | 29.0 | 15.5 | 2.7 | 11.1 |
| | No Home Reported | 5.9 | 23.9 | 16.5 | 70.2 |
| MedicaidS | Yes | 23.8 | 48.4 | 8.3 | 3.3 |
| UniqueAgents | 4 | 11.6 | 2.2 | 8.6 | – |
| | 3 | 27.9 | 31.9 | 22.3 | 2.3 |
| | 2 | 60.6 | 65.9 | 69.1 | 97.7 |
| FinalProgPhase | Level 3/4 | 11.1 | 15.7 | 32.3 | 0.3 |
| | Level 1/2 | 56.5 | 14.4 | 22.7 | 3.1 |
| | Level 0 | 2.9 | 35.5 | 7.0 | 27.0 |
| RewardedBehv | Yes | 4.0 | 29.1 | 2.5 | 1.5 |
| Sanctions | Yes | 91.8 | 99.3 | 89.8 | 41.1 |

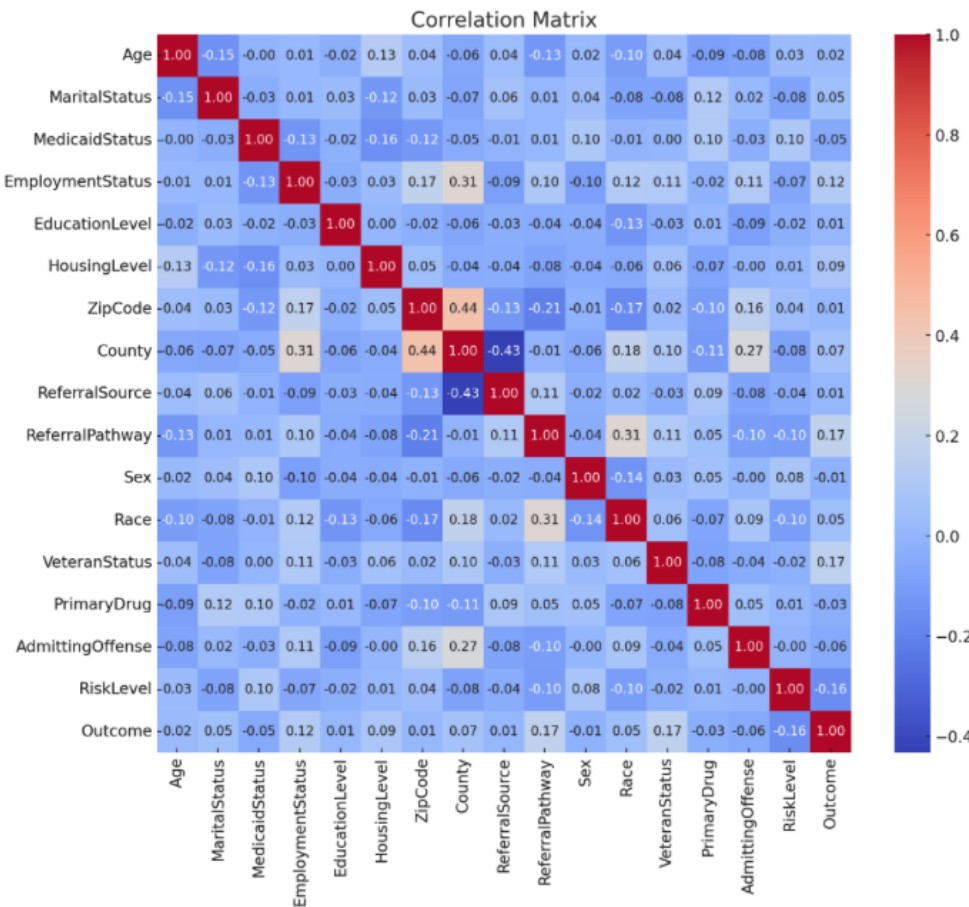

Figure 10: Correlation Matrix of features

