# OpenReview forum: "Latent Feature Mining with Large Language Models"
_NeurIPS.cc/2024/Conference — Submitted to NeurIPS 2024_

### Official Review · Reviewer_imv4 · 2024-07-05

**Soundness:** 3
**Presentation:** 3
**Contribution:** 3
**Rating:** 5
**Confidence:** 3

**Summary:**

This paper proposes a framework to augment latent features from observed features, with the help of LLM. They frame the problem as a text-to-text reasoning problem.  The method can be adapted to different domains easily.  The method is also validated with a real world dataset.

**Strengths:**

Overall, the presentation and logic flow are smooth and clear.  The methodology is also reasonable to me. And their experiments also validate the effectiveness of their method.

**Weaknesses:**

The key concern for me is that when using the LLM for inference and text generation, I worry about the social bias and fairness of the problem. Some research has shown that LLM is still biased in some sense, can the author conduct some evaluation on whether the latent feature is biased towards some sensitive attributes like race, gender, etc?

The other thing is that I wonder how much human labor effort and expert labor effort will be needed to have the latent features.

Typo in line 203

**Questions:**

In line 166, how do you determine the number l ?

---

> ### Author Rebuttal · Authors · 2024-08-07
>
> **Thank you for your comments and feedback. Here are our responses to your questions:**
>
> ---
>
>  - **Question 1: Suggestion on adding experiments about social bias and fairness of using LLMs for inference.**
>  - **Response:** We appreciate your concern about the bias and fairness of using LLMs for inference. Following your suggestion, we conduct **additional experiments to validate the LLMs’ inherent bias is not carried into the inference process.**
> 	 -   ***Experiment setting***: Determine if the reasoning process within generated texts exhibits biases related to social-economic status or risk assessment, specifically racial biases.
> 	 -   ***Experiment results:***
> 			- For social-economic status: We implemented a pretrained keyword extraction model YAKE [1] to search for racial terms in the reasoning steps of the text, with results indicating that such keywords were not found, suggesting **no explicit racial bias in this context**.
> 			-  For risk level assessment: we closely examined the race distribution in the ground-truth data versus the distribution in the predictions made by the model. The analysis revealed that the race distributions between the ground-truth and the predicted outcomes are similar. This similarity suggests that **the model does not introduce additional racial biases in its predictions** and reflects the distributions present in the input data accurately.
>
> ---
>
> - **Question 2:  How much human labor effort and expert labor effort will be needed to have the latent features?**
> - **Response:** To obtain latent features, there are two stages that need human expert input.
> 	 - **The first stage involves developing foundational guidelines** (i.e., the standard solutions to guide LLMs). We propose a crafting rationales strategy to enhance the efficiency of this step, which simplifies the creation of baseline rationales, ensuring that the framework for latent feature extraction is both robust and effectively grounded (*please see page 6, under "In the second sub-step...*). This stage only requires expert knowledge, **minimum labor.**
>      Moreover, the expert input required is not labor-intensive but rather experience-based. For instance, we successfully applied the same framework to a different domain (healthcare) in a short timeframe, indicating that the process of adapting the trained LLMs to new domains is efficient and not overly demanding in terms of labor.
>
> 	 - **The second stage requiring human involvement is the validation phase**, which occurs as step 2 in our process. Although this stage has been designed to be as streamlined as possible, it still requires human oversight to ensure the accuracy and relevance of the features being extracted. However, this process has been optimized to **minimize labor**, focusing on quality control without demanding extensive time from our experts.
>
>
>   We note that **our framework requires significantly less than what is traditionally required in human annotation approaches** for latent feature mining, yet this efficiency does not compromise the quality of the outcomes.
>
> ---
>
> - **Question 3: How to infer the first intermediate predicate ?**
> - **Response:** From our interpretation of your question, we assume that you are asking how the first intermediate predicate (P_1) be inferred. By leveraging domain expertise, we formulate initial hypotheses or predicates about potential relationships between features. This relates to our above response to your question 2, where the first stage of our framework requires human expert opinion to develop foundational guidelines (i.e., the standard solutions to guide LLMs).
>
> ---
>
> **We appreciate your comments and questions, and hope this response addresses your questions. We look forward to any further feedback you may have.**
>
> ----
>
> [1] Campos, R., Mangaravite, V., Pasquali, A., Jatowt, A., Jorge, A., Nunes, C. and Jatowt, A. (2020). YAKE! Keyword Extraction from Single Documents using Multiple Local Features. In Information Sciences Journal. Elsevier, Vol 509, pp 257-289.

---

> > ### Comment · Reviewer_imv4 · 2024-08-13
> >
> > Thank you for the thoughtful rebuttal.  I will be maintaining my current score.

---

### Official Review · Reviewer_QvzM · 2024-07-09

**Soundness:** 2
**Presentation:** 2
**Contribution:** 2
**Rating:** 3
**Confidence:** 3

**Summary:**

The authors propose a unique form of LLM data-augmentation that attempts to generate informative latent variables to improve downstream tasks. They do this by transforming the latent feature mining task into a text-to-text propositional reasoning task.
Validation is performed with a case study in the criminal justice system and latent features align well with ground truth labels + significantly enhance downstream classifier performance.

**Strengths:**

- Clear, well written paper with descriptive diagrams
- Using LLMs to infer latent space in this way seems to be a novel idea

**Weaknesses:**

- Type on line 203: "whic serve"
- Potential for LLM biases in the latent variable finding. E.g. Marijuana usage does not necessarily require Substance Abuse Treatment.
- Lack of evaluation on multiple datasets/domains and no publically released code
- Lack of ablations exploring generalizability with x\% features removed

**Questions:**

See weakness

- How does this compare with other forms of data augmentation?
- How does this compare with the traditional methods of obtaining latent space representations -> VAE, etc?

**Limitations:**

Yes

---

> ### Author Rebuttal · Authors · 2024-08-07
>
> **Thank you for your valuable feedbacks and comments, here are our response to your questions:**
> - **Question1: Concern about potential biases from LLMs in latent variable finding.**
> - **Response:** Thank you for highlighting this concern. **Our framework aims to minimize biases by leveraging domain-specific data and expert input during the fine-tuning processes.** We carefully design rules based on domain knowledge and expert input. For example, the client shown in Figure 1 primarily uses marijuana but has a property offense as the admitting offense. Our community partner noted that marijuana usage often correlates with other substance use (though not recorded in the admitting offense) and that “the prevention of heavy marijuana use could potentially reduce property crime in the future” [1]. Therefore, LLMs, following expert guidance, add substance abuse treatment to the potential requirement list.
>
> 	As addressed in another comment by Reviewer U9i4, we conducted a thorough sanity check, showing that **LLMs trained in our framework do not amplify bias but adhere to domain expert principles** (the “standard solutions” provided for training).
>
> 	While bias may be inherent in input data, **our framework ensures transparency in inference and prediction, aiding result interpretation and bias correction through human-in-the-loop processes**, which allows domain experts to adjust and refine the model’s reasoning pathways. Consequently, inferred latent features align with nuanced real-world outcomes rather than broad statistical correlations.
> - **Question 2: Can you add experiments on other domains/dataset to prove generalizability.**
> - **Response:** We have conducted additional experiments in the healthcare domain with the MIMIC-IV dataset to prove the generalizability of our framework. Here are the experiment details and results:
> 	- **Dataset:** The MIMIC dataset is a comprehensive dataset containing detailed de-identified patient clinical data and is widely used for various prediction tasks in the machine learning literature.
> 	- **Task Description:** The discharge location prediction task uses patient data to forecast the most likely discharge destination, aiding hospital management in preparing for discharges. We leverage our framework to introduce a "social support" feature, which captures the healthcare and community support available to the patient. This involves repeating the four-step process of our framework:
> 		- ***Step 1.*** Create rationale: we leverage domain expertise in hospital inpatient management and patient flow to help us craft rationales to infer social support.
> 		- ***Step 2.*** Enlarge synthetic data for LLM training: similar to our approach for the outcome prediction task using the criminal justice data (task 2 mentioned in the main paper), we use GPT-4 to generate 4500 data points to fine-tune GPT-4o-mini.
> 		- ***Step 3 & 4.*** The remaining two steps are the same as those used for the outcome prediction task (task 2 mentioned in the main paper).
> 	- **Experiment Setting:** Due tack the ground-truth label for the latent variable "social support," making our experiments akin to Task 2 in the main paper. After generating the latent features, we train four machine learning classifiers: Logistic Regression (LR), Multilayer Perceptron (MLP), Gradient Boosting Trees (GBT), and Random Forest (RF). Training is conducted with and without latent features for comparison. Each experiment is run five times with different random seeds, and the results are averaged to ensure reliability.
> 	- **Experiment Results:** The table below demonstrates the experiment results, showing an average improvement of approximately 8.64% in accuracy and 8.64% in F1 score when latent features are added to the models. This is similar to the percentage increase reported in Table 4A in the main paper.
>
> 	| Model | Accuracy (std.) | F1_score (std.) |
> 	|:----------:|:-----------------:|:-----------------:|
> 	| LR | 65.22% (0.01) | 65.46% (0.01) |
> 	| MLP | 63.19% (0.02) | 63.19% (0.02) |
> 	| GBT | 64.84% (0.01) | 65.09% (0.01) |
> 	| RF | 65.11% (0.01) | 65.44% (0.01) |
> 	| LR w/ LF | **71.22% (0.01)** | **71.26% (0.01)** |
> 	| MLP w/ LF | **74.40% (0.01)** | **74.50% (0.01)** |
> 	| GBT w/ LF | **75.56% (0.02)** | **75.38% (0.02)** |
> 	| RF w/ LF | **75.31% (0.01)** | **75.22% (0.01)** |
>
> 	The results demonstrate **another strong evidence of using our framework to improve downstream prediction power with the addition of latent features**. Connecting to Lemma 1, our experiment results show that the added features are informative – likely because the human experts (case managers or physicians) are making decisions based on more than the explicitly recorded features (X) in the dataset.
> - **Question 3: How does this compare with other forms of data augmentation?**
> - **Response:** Our framework leverages LLMs to augment observed features in datasets with latent features. This approach differs from traditional data augmentation methods like VAE, as we augment the dimensions of X rather than the sample sizes. A detailed comparison: VAEs aim to represent the original feature space in a lower-dimensional space, whereas our goal is to add mined features to enhance prediction power for downstream tasks. VAEs use probabilistic approaches to describe data distribution with latent variables, but their mappings can be difficult to interpret, while our framework offers better interpretability. Section 2 contains more detail comparison with other methods.
>
> ----
>
> **We sincerely thank you for your feedback and comments, and hope this response addresses your question. We are looking forward to further discussion. If our responses have addressed your concerns, we kindly request a reconsideration of the rating score. Thank you again for your valuable input!**
>
> ----
>
> [1] Green KM, etc al. Does heavy adolescent marijuana use lead to criminal involvement in adulthood? _Drug Alcohol Depend_.  doi:10.1016/j.drugalcdep.2010.05.018

---

> ### Author Response · Authors · 2024-08-07
>
> **Thank you for your  comments and valuable suggestions, here are our response to your suggestions:**
>
> ---
>
> - **Suggestion1:  Add ablations exploring generalizability with x% features removed.**
> - **Response:** Thank you for your valuable suggestion regarding the inclusion of ablation studies. We recognize this as an important aspect of framework evaluation and have thus conducted additional experiments to gauge the robustness and dependence of our model on specific features. We systematically removed different proportions of features and reran experiments on the risk level prediction task.
> 	- **Experiment 1: Removing Features Mentioned in the Provided Guidelines**
>   As discussed in the paper, we use domain knowledge and expert input to provide guiding principles for the LLMs' inference process. For this experiment, we removed two features explicitly mentioned in the guiding principles—age and marriage status—and observed the following impacts on model performance:
>
>     | Model | ROC_AUC score |
>     |:------------------:|:-------------:|
>     | MLP | 50% |
>     | GBT | 43% |
>     | GPT3.5 Baseline | 56% |
>     | GPT3.5 Full-feature | 75% |
>
>   The results demonstrate that excluding features specified in the guiding principles (e.g., GPT3.5 Baseline) reduces the final accuracy but still outperforms traditional machine learning models such as MLP and Gradient Boosting Trees (GBT).
>
> 	- **Experiment 2: Removing Features Not Included in the Provided Guidelines**
>   In this second experiment, we removed the feature “referral source”, which was not explicitly recommended to be used in the guiding principles:
>
>   | Model | ROC_AUC score |
>   |:------------------:|:-------------:|
>   | MLP | 54% |
>   | GBT | 58% |
>   | GPT3.5 Baseline | 71% |
>   | GPT3.5 Full-feature | 75% |
>
> These results indicate that the exclusion of features not included in the guiding principles (e.g., GPT3.5 Baseline) only slightly reduces final accuracy, yet continues to outperform the machine learning baselines (MLP and GBT).
>
>   ---
>
>  - **Suggestion 2:**  Release code for more comprehensive evaluation
> - **Response:** We planed to release the code after the review. Following your suggestion, we are providing access to the code for the MIMIC experiments as a preview. Following the rebuttal instruction, we have submitted an anonymous link to the code to the AC. This link will become available for preview upon approval by the AC.
>
> ----
>
> **Thank you again for your valuable input! We are looking forward to further discussion.**

---

> > ### Comment · Reviewer_QvzM · 2024-08-09
> >
> > The additional evaluations are more encouraging, especially the ablations with Removing Features Mentioned in the Provided Guidelines as well as the Removing Features Not Included in the Provided Guidelines. However, I still think it is quite difficult to validate the quality of the model even with the MIMIC dataset, as the interpretability results are still quite difficult to evaluate in its current state. Even if human-in-the-loop evaluations are used on the generated examples, there is no guarantee of the trend holding for different datasets, especially those out of distribution (e.g. LLM's reasoning on highly specific domains such as chemistry remains quite nonsensical).
> >
> > I am not entirely convinced by the following responses to Reviewer U9i4's first or second points. From personal experience, "logical reasoning leads to correct predictions, and incorrect reasoning does not produce correct results" seems to slightly contradict the results found in [1], where COT explanations can be plausible yet misleading. Also, any LLM that "extrapolates information not explicitly present in the original data" could also suffer from hallucinations, which is an open problem.
> >
> > I'd like to thank the authors for putting in the time to answer my responses, but I still think it's difficult for me personally to accept the paper.
> >
> > [1] Turpin, M., Michael, J., Perez, E., & Bowman, S. (2024). Language models don't always say what they think: unfaithful explanations in chain-of-thought prompting. Advances in Neural Information Processing Systems, 36.

---

> > > ### Author Response · Authors · 2024-08-10
> > >
> > > **Thank you for taking the time to review our work and for replying our responses.** We appreciate the opportunity to clarify and expand on our findings. Here are our response to your concerns **point-by-point**:
> > >
> > > ---
> > >
> > > ***Concern on generalizability:***
> > >
> > > > "It is quite difficult to validate the quality of the model even with
> > > > the MIMIC dataset. Even if human-in-the-loop evaluations are used on
> > > > the generated examples, there is no guarantee of the trend holding for
> > > > different datasets.”
> > >
> > > As discussed in Sections 1 and 4 of our paper, our framework is designed to **enhances Machine Learning model by externalizing expert knowledge to generate new features**. Our experiments with the MIMIC dataset demonstrate that, with sufficient expert input, **our framework generates strong features that significantly enhance predictive accuracy**. The success observed in the two tested domains indicates that our framework is **highly flexible and can be effectively transferred to other domains**.It is important to note that adapting it to new areas requires careful integration of expert knowledge as a crucial first step to ensure that the framework maintains its robustness and accuracy across different applications
> > >
> > >   ---
> > >
> > > ***Concern on the robustness of CoT :***
> > >
> > > > "From personal experience, 'logical reasoning leads to correct
> > > > predictions, and incorrect reasoning does not produce correct results
> > > > seems to slightly contradict the results found in. COT
> > > > explanations can be plausible yet misleading.“
> > >
> > > **Thank you for highlighting this concern.** We carefully reviewed the paper you cited. Their experiments are conducted in **zero-shot and few-shot settings**. We agree that CoT can be plausible generate misleading or biased output in few shot setting. It’s important to note that our framework trains LLMs to better align with human knowledge. Fine-tuning, as an important component of our approach, significantly increase the quality of generated CoT, and enhances the reasoning capabilities of LLMs. As indicated in Section 6 (line 370) of our paper, our ablation study on the **fine-tuning process shows that fine-tuned LLMs substantially outperform those in zero-shot and few-shot scenarios**.
> > >
> > > Additionally, **CoT has been validated as an effective technique across various domains**. For instance, the CoT strategy has been shown to significantly improve LLMs' performance in document understanding and citation generation[1]. It also helps VLMs mimic multi-hop reasoning in answering SCIENCEQA questions [2]. Additionally, LLMs fine-tuned with CoT exhibit marked improvements in reasoning ability across different datasets [3].
> > >
> > >  ---
> > >
> > >  ***Concern on the robustness of Hallucination :***
> > >
> > > > "Any LLM that extrapolates information not explicitly present in the
> > > > original data could also suffer from hallucinations, which is an open
> > > > problem.”
> > >
> > > **We address hallucinations through fine-tuning and post-generation validation, aiming to filter out such inaccuracies**. Moreover, our error analysis of the generated reasoning text revealed that the fine-tuned LLMs consistently used accurate information from the profile without fabricating details  (For more details, please refer to our response to Reviewer U9i4). **As you mentioned,  hallucination remains an open problem in the field, it falls beyond the immediate scope of our work**. We will leave this issue as a direction for future research.
> > >
> > > Additionally, figure 3 in the paper presents the fine-tuned LLMs' inference results on risk level prediction task, where **our framework was applied to infer features with known ground truth**. The results show that LLMs is able to high-quality feature generation even when extrapolating to unseen information, and outperform traditional Machine Learning approaches in accurately inferring latent features. This demonstrates the accuracy and effectiveness of our framework, giving us confidence in **its ability to generate high-quality features**.
> > >
> > > ---
> > >
> > > **Thanks for engaging the discussion period. We are looking forward any further feedback. If this response address your concerns, we kindly request a reconsideration of our merits. Thank you again for your valuable input!**
> > >
> > > ---
> > >
> > > ***References:***
> > >
> > > [1] Ji, B., Liu, H., Du, M., & Ng, S.-K. (2024). Chain-of-Thought Improves Text Generation with Citations in Large Language Models. Proceedings of the AAAI Conference on Artificial Intelligence, 38(16), 18345-18353. https://doi.org/10.1609/aaai.v38i16.29794
> > >
> > > [2] Lu, P., Mishra, S., Xia, T., Qiu, L., Chang, K. W., Zhu, S. C., ... & Kalyan, A. (2022). Learn to Explain: Multimodal Reasoning via Thought Chains for Science Question Answering. Advances in Neural Information Processing Systems, 35, 2507-2521.
> > >
> > > [3] Ho, N., Schmid, L., & Yun, S. Y. (2023, July). Large Language Models Are Reasoning Teachers. Proceedings of the 61st Annual Meeting of the Association for Computational Linguistics (Volume 1: Long Papers), 14852-14882.

---

### Official Review · Reviewer_U9i4 · 2024-07-11

**Soundness:** 3
**Presentation:** 3
**Contribution:** 2
**Rating:** 4
**Confidence:** 4

**Summary:**

The paper presents a framework that uses LLMs to improve predictive modeling by augmenting observed features with inferred latent features. This approach transforms the latent feature mining task into a text-to-text propositional reasoning task, enabling LLMs to infer unobserved yet crucial factors from available data. The framework is tested through a case study in the criminal justice system, demonstrating improved accuracy in scenarios where collected features are weakly correlated with outcomes.

**Strengths:**

1. It addresses the challenge of limited data availability by leveraging LLMs to infer latent features, to improve predictive modeling.

2. The approach of transforming latent feature mining into a text-to-text propositional reasoning task is interesting.

3. The validation on criminal justice data, shows potential for broader applications. The method's generalizability across different domains with minimal customization is a significant advantage, and the reduced need for extensive human-annotated training data makes it practical and scalable.

**Weaknesses:**

1. The paper does not adequately address how to measure the impact of errors introduced by the LLM-based solution on predicted outcomes, nor does it provide uncertainty estimates. This process surely introduces errors, as with any ML-based solution. How do we measure its effect on predicted outcomes, including uncertainty estimates? I suspect more labeled data would be needed to assess this properly (see Egami et al. @ NeurIPS 2023).

2. It is unclear whether the approach should be viewed as a form of dimensionality reduction (based on existing features) or if it extrapolates information not present in the original data. This ambiguity raises concerns about potential bias amplification. For instance, Figure 1 shows deductions made by the model that are not clearly supported by the evidence, suggesting that the method might be amplifying existing biases rather than mitigating them.

3. Connecting to the previous point, the method's ability to learn latent information that is causally predictive of the outcome, as opposed to relying on spurious correlations, remains uncertain. Conducting an out-of-distribution test, where the characteristics of individuals differ from the training data, would be crucial in evaluating the model's generalizability and causal inference capabilities.

4. The rationale behind not using all available data directly in the LLM for prediction is not well-justified. Directly prompting the LLM with the full data might provide more accurate predictions without the need for dimensionality reduction.

**Questions:**

1. How do you measure the impact of errors introduced by the LLM-based solution on predicted outcomes, including uncertainty estimates?

2. Should the proposed method be viewed as a form of dimensionality reduction, or does it extrapolate information not present in the original data? How do you ensure that this process does not amplify existing biases?

3. How do you ensure that the method learns latent information that is causally predictive of the outcome and does not simply rely on spurious correlations?

**Limitations:**

The authors have acknowledged limitations of their work, particularly in addressing the ethical concerns associated with data collection and the need for privacy.

---

> ### Author Rebuttal · Authors · 2024-08-07
>
> **Thank you for your valuable feedbacks!**
> -   **Question 1: How to measure the impact of errors? How to ensure the approach doesn't amplify potential errors?**
> -   **Response:** As discussed in Section 4, we incorporate human-in-the-loop interventions to identify and remove erroneous synthetic reasoning steps from the training data. Additionally, we utilize automatic evaluation metrics to detect errors within the reasoning process and excluded from the training dataset to maintain model integrity.
> 	Furthermore, we conducted an additional error analysis for the Risk Level Prediction (Task 1 in the main paper) to evaluate the impact of errors. We recruit human volunteers examine all 1168 instances used in Step 2 (generating synthetic rationales) of the framework. Our analysis found that logical reasoning leads to correct predictions, and incorrect reasoning does not produce correct results, thereby minimizing error amplification. This analysis supports the effectiveness of our framework in avoiding error amplification.
> -   **Question 2: Clarification on the difference with dimension reduction.**
> -   **Response:** Our method extrapolates information not explicitly present in the original data, unlike dimension reduction techniques like VAE or PCA, which represent the original feature space in lower dimensions while retaining as much information as possible. Instead, our framework enriches the dataset by adding mined features to improve prediction power for downstream tasks, mimicking human experts' reasoning by considering features holistically and inferring additional socio-economic information not recorded in the data. Figure 1 in our paper illustrates this step-by-step extrapolation process using LLMs.
> - **Question 3: Suggestion on adding experiments on other domains/dataset to prove generalizability.**
> - **Response:** We have conducted additional experiments in the healthcare domain with the MIMIC-IV dataset to prove the generalizability of our framework. Here are the experiment details and results:
> 	- **Dataset:** The MIMIC dataset is a comprehensive dataset containing detailed de-identified patient clinical data and is widely used for various prediction tasks in the machine learning literature.
> 	- **Task Description:** The discharge location prediction task uses patient data to forecast the most likely discharge destination, aiding hospital management in preparing for discharges. Our latent feature framework creates features that enhance machine learning models for this task. Notably, we introduce a "social support" feature, which captures the healthcare, familial, and community support available to the patient. This involves repeating the four-step process of our framework:
> 		- ***Step 1.*** Create rationale: we leverage domain expertise in hospital inpatient management and patient flow to help us craft rationales to infer social support.
> 		- ***Step 2.*** Enlarge synthetic data for LLM training: similar to our approach for the outcome prediction task using the criminal justice data (task 2 mentioned in the main paper), we prompt GPT-4 to generate 4500 data points to fine-tune GPT-4o-mini.
> 		- ***Step 3 & 4.*** The remaining two steps are the same as those used for the outcome prediction task.
> 	- **Experiment Setting:** Due to the lack the ground-truth label for the latent variable "social support," making our experiments akin to Task 2 in the main paper. After generating the latent features, we train four machine learning classifiers: Logistic Regression (LR), Multilayer Perceptron (MLP), Gradient Boosting Trees (GBT), and Random Forest (RF). Training is conducted with and without latent features for comparison.  Each experiment is run five times with different random seeds, and the results are averaged to ensure reliability..
> 	- **Experiment Results:** The table below demonstrates the experiment results, showing an average improvement of approximately 8.64% in accuracy and 8.64% in F1 score when latent features are added to the models. This is similar to the percentage increase reported in Table 4A in the main paper.
> 	| Model | Accuracy (std.) | F1_score (std.) |
> 	|:----------:|:-----------------:|:-----------------:|
> 	| LR | 65.22% (0.01) | 65.46% (0.01) |
> 	| MLP | 63.19% (0.02) | 63.19% (0.02) |
> 	| GBT | 64.84% (0.01) | 65.09% (0.01) |
> 	| RF | 65.11% (0.01) | 65.44% (0.01) |
> 	| LR w/ LF | **71.22% (0.01)** | **71.26% (0.01)** |
> 	| MLP w/ LF | **74.40% (0.01)** | **74.50% (0.01)** |
> 	| GBT w/ LF | **75.56% (0.02)** | **75.38% (0.02)** |
> 	| RF w/ LF | **75.31% (0.01)** | **75.22% (0.01)** |
>
> 	This experiment on a different dataset from a different domain shows the effectiveness and generalizability of our framework. Connecting to Lemma 1, our experiment results show that the added features are informative – likely because the human experts (case managers or physicians) are making decisions based on more than the explicitly recorded features (X) in the dataset.
>
> -   **Question 4: How to ensure that the method learns latent information that is causally predictive of the outcome ?**
> -   **Response:** We acknowledge that our current framework does not explicitly perform causal inference. Instead, relationships between features and outcomes are identified through domain knowledge and expert input, leveraging expert understanding to guide the identification of meaningful latent features. We agree on the importance of causally relevant latent features and recognize this as a crucial direction for future research. However, incorporating causal inference into our framework is beyond the scope of this paper and would require a dedicated study to thoroughly examine and address this issue.
>
> ----
>
> **We sincerely thank you for your feedback and comments, and hope this response addresses your question. We look forward to further feedbacks. If our responses have addressed your concerns, we kindly request a reconsideration of the rating score. Thank you again for your valuable input!**

---

> > ### Comment · Reviewer_U9i4 · 2024-08-12
> >
> > thank you for thoughtfully addressing my questions, and especially for adding an experiment.
> >
> > i still think that a more careful design w.r.t to causality and potential errors would make this paper substantially more useful.

---

### Official Review · Reviewer_bSc5 · 2024-07-18

**Soundness:** 2
**Presentation:** 3
**Contribution:** 2
**Rating:** 6
**Confidence:** 4

**Summary:**

This paper used large language models to infer latent variables that are important for downstream prediction tasks to augment the existing models. In particular, the author demonstrated the use of the proposal on a criminal justice system use case, in which the LLM-mined-latent features significantly boost the prediction performance.

Overall the paper presents an interesting question and how LLM could help mining for latent features, but I have several questions regarding 1. the generalizability of the proposed method; 2. Appropriate combinations/baseline methods; and 3. The potential ethical implications of this method. I will detail these points in the strength/weakness sections below.

**Strengths:**

1. On the outcome prediction results, using the latent feature seems to have boosted the performance by 7-10%, a big margin.
2. The proposed framework brings a level of formalism to the current crowded LLM for (social) science applications/work, including the text-to-text proposition work.

**Weaknesses:**

1. Despite the general initial framing in Section 3, it was not very clear how generalizable results from Sections 4-6 are — this includes not only the COT and the prompts used, but more critically the selection of what kind of latent features we are including.
2. The current work does not seem to go into depth about what kind of latent features are LLM particularly good at constructing and which ones are particularly “bad” (eg subject to the most bias and systematic over-or-under-prediction). I think this is particularly relevant for social science applications where many of the categories and features are more of a “construct” and often qualitative in nature.
3. Have the authors compared the results by using text embeddings of the descriptions as an input feature? It’s interesting that the fine-tuning strategy is necessary for good performance, which seems to suggest that learning the intermediate classification rule is important since direct manipulation of natural language yields less impressive  results.

**Questions:**

1. In addition to the questions listed in the weakness section, I would encourage the authors to engage much more critically with the limitations of these approaches, esp. when high-risk predictions in CJS are very high-stake.
2. I think this paper would also be much stronger if the proposed framework of mining for latent features (which requires quite extensive human rationale collection) works for another prediction problem (either different data and/or different tasks).

**Limitations:**

Mentioned above.

---

> ### Author Rebuttal · Authors · 2024-08-07
>
> **Thank you for your comments and feedbacks. Here are our responses to your questions:**
>
> ---
>
>  **Question 1: Suggestion on adding experiments on other domains/dataset to prove generalizability.**
>
> **Response:**
> We have conducted additional experiments in the healthcare domain with the MIMIC-IV dataset to prove the generalizability of our framework. Here are the experiment details and results:
>
> - **Dataset:** The MIMIC (Medical Information Mart for Intensive Care) dataset [2] is a comprehensive dataset containing detailed de-identified patient clinical data and is widely used for various prediction tasks in the machine learning literature.
>
> - **Task Description:** The discharge location prediction task uses patient data to predict the most likely discharge destination, aiding hospital management in preparing for discharges. We apply our latent feature framework to create features that enhance machine learning models for this task. Specifically, we introduce a "social support" feature, capturing healthcare, familial, and community support available to the patient. We repeat the four-step process of our framework:
>
> 	- ***Step 1.*** Create rationale: we leverage domain expertise in hospital inpatient management and patient flow to help us craft rationales to infer social support.
>
> 	- ***Step 2.*** Enlarge synthetic data for LLM training: similar to our approach for the outcome prediction task using the criminal justice data (task 2 mentioned in the main paper), we use a self-instructing approach to prompt GPT-4 to generate 4500 data points to fine-tune GPT-4o-mini.
>
> 	- ***Step 3 & 4.*** The remaining two steps are the same as those used for the outcome prediction task (task 2 mentioned in the main paper).
>
> - **Experiment Setting:** Note that we do not have the ground-truth label for the latent variable "social support," making our experiments similar to Task 2 in the main paper. After generating the latent features, we train four machine learning classifiers: Logistic Regression (LR), Multilayer Perceptron (MLP), Gradient Boosting Trees (GBT), and Random Forest (RF). We conduct training with and without latent features for comparison. The dataset is split 70/30 for training and testing. Each experiment is run five times using five different random seeds, averaging the results to ensure reliability.
>
> - **Experiment Results:** The table below demonstrates the experiment results, showing an average improvement of approximately 8.64% in accuracy and 8.64% in F1 score when latent features are added to the models. This is similar to the percentage increase reported in Table 4A in the main paper.
>
> 	| Model | Accuracy (std.) | F1_score (std.) |
> 	|:----------:|:-----------------:|:-----------------:|
> 	| LR | 65.22% (0.01) | 65.46% (0.01) |
> 	| MLP | 63.19% (0.02) | 63.19% (0.02) |
> 	| GBT | 64.84% (0.01) | 65.09% (0.01) |
> 	| RF | 65.11% (0.01) | 65.44% (0.01) |
> 	| LR w/ LF | **71.22% (0.01)** | **71.26% (0.01)** |
> 	| MLP w/ LF | **74.40% (0.01)** | **74.50% (0.01)** |
> 	| GBT w/ LF | **75.56% (0.02)** | **75.38% (0.02)** |
> 	| RF w/ LF | **75.31% (0.01)** | **75.22% (0.01)** |
>
> 	The results demonstrate **another strong evidence of using our framework to improve downstream prediction power with the addition of latent features**. This experiment on a different dataset from a different domain shows the effectiveness and generalizability of our framework. Connecting to Lemma 1, our experiment results show that the added features are informative – likely because the human experts (case managers or physicians) are making decisions based on more than the explicitly recorded features (X) in the dataset.
>
> ---
>
> **Question 2: Suggestion on using text embedding to show the effectiveness of learning**
>
> **Response:** -   Thank you for this interesting suggestion. From our interpretation of your comment, we assume that you are asking if we have compared our results with an alternative approach where text embeddings (numerical representations of text) of the descriptions are used as input features directly. We make two clarifications. First, the given input features X from our datasets are either continuous (numeric) or categorical (discrete) numbers. There is no natural text embedding directly available from the data. Thus, we use the Chain of Thoughts (CoT) prompting as a critical component of our framework, to provide the interpretable text input after processing the numeric features X. Hence, there is no direct alternative to compare if you were thinking of replace the input features with text embeddings.
>
> Second, integrating text embeddings after profile writing is an idea worth exploring. We acknowledge that it could be an interesting alternative and warrants further exploration. Meanwhile, we note that the replacement of natural language to text embeddings as input could obscure the step-by-step logical flow that is essential for the transparency and interpretability of the CoT process.  Nevertheless, though we have not specifically compared this approach in our current study, we recognize that it could provide valuable insights. Future work could explore integrating text embeddings alongside our fine-tuning strategy to potentially combine the strengths of both methods. Our conjecture is that fine-tuning strategy is necessary for achieving good performance. This aligns with your suggestion that the process of learning an intermediate classification rule during fine-tuning is crucial for the model’s success.
>
> ----
>
> **Thank you for your insightful comments and questions. We hope our response has been helpful in addressing your concerns. We are looking forward to further discussion.**
>
> ----
>
> [1] Johnson, A. E. W., Pollard, T. J., Shen, L., Li-Wei, H. L., Feng, M., Ghassemi, M., ... & Mark, R. G. (2016). MIMIC-III, a freely accessible critical care database. Scientific Data, 3, 160035.

---

> ### Author Response · Authors · 2024-08-07
>
> **Thank you for  thoughtful comment! Following your suggestion, we plan to consider adding following discussions into the main paper.**
>
> ---
>
> - **Discussion 1: What are limitations of the framework when high-risk predictions in CJS are very high-stakes?**
> -   **Response:** -   Thank you for this insightful question. Below we list more detailed discussion on the limitations of the framework for high-risk predictions in the criminal justice system (CJS). We will incorporate this discussion into the main paper during our revision process.
>
> 	- **Data Quality and Dependency:** The performance and reliability of the framework depend on the quality of both the training data and the input data used during operation. Inconsistencies, errors, or gaps in data can lead to inaccurate predictions, which is critical when these predictions aim to influence decisions about individuals’ freedoms in the CJS. We are cautious of such errors and this is why we closely involve human experts during the whole research process. While acknowledging that bias sometimes is inherent in the input data, we emphasize that our framework provides the transparency of how the inference/prediction is made, which can significantly help interpret results and correct for such bias by introducing human-in-the-loop. In other words, our framework allows an iterative fine-tuning process, where we can incorporate feedback loops with domain experts to adjust and correct the model’s reasoning pathways. This iterative method ensures that the latent features inferred, such as the need for substance abuse treatment, are aligned with nuanced real-world outcomes rather than broad statistical correlations.
>
> 	- **Impact on Public Trust:** The use of AI in areas impacting fundamental rights can affect public trust in the justice system. If the public perceives these tools as opaque or biased, it could undermine confidence in judicial processes, which is critical for the effective functioning of any democratic legal system. LLMs provide the interpretability that is useful for gaining public trust and allows an iterative process to refine the prediction results.
>
> ---
>
> -   **Discussion 2: What kind of latent features are your framework particularly good at constructing ?**
> -   **Response:** Thank you for such an insightful question. Our framework excels at constructing latent features from observable and structured data with clear relationships, such as socio-economic status indicators and risk levels. These features benefit from well-defined guidelines and domain knowledge, allowing our framework to infer them with higher accuracy and reliability by mimicking the human reasoning process.
>
> 	The framework can be less effective with subjective latent features or those with subtle relationships, like personal behavioral tendencies. These complexities can introduce bias and lead to overfitting and systematic errors. To mitigate these issues, we use domain expertise and a human-in-the-loop approach to refine latent feature construction. Despite these measures, some biases may persist, and addressing them remains a focus for future research. We will continue to explore improvements.
> ----
>
> **Thank you again for your valuable input! We are looking forward to further discussion.**

---

### Decision · Program_Chairs · 2024-09-25

**Decision:**

Reject

**Comment:**

The paper proposes a framework of using LLMs for reasoning in a chain-of-thought fashion. The framework is applied to a real world problem on criminal justice data. The following weaknesses were raised:
- The approach is tested only for a specific application (criminal justice). Although in the rebuttal, the authors came up with a second application, it seems that significant manual effort is required to generalize the approach to each new application.
- The use of LLM to generate synthetic data for such applications raises the problem of biases in the LLMs. This is especially important in the application that is considered in this paper. Although manual examples are provided to guide the LLM, there is insufficient discussion on how the LLM could be checked against biases.